# On the Conditioning Consistency Gap in Conditional Neural Processes

**Robin Young**  *robin.young@cl.cam.ac.uk*
*Department of Computer Science and Technology*
*University of Cambridge*

**Reviewed on OpenReview:** *https://openreview.net/forum?id=rLJ5Hm5vbG*

## Abstract

Neural processes are meta-learning models that map context sets to predictive distributions. While inspired by stochastic processes, NPs do not generally satisfy the Kolmogorov consistency conditions required to define a valid stochastic process. This inconsistency is widely acknowledged but poorly understood. Practitioners note that NPs work well despite the violation, without quantifying what this means. We address this gap by defining the conditioning consistency gap, a KL divergence measuring how much a conditional neural process's (CNP) predictions change when a point is added to the context versus conditioned upon. Our main results show that for CNPs with bounded encoders and Lipschitz decoders, the consistency gap is $O(1/n^2)$ in context size $n$, and that this rate is tight. These bounds establish the precise sense in which CNPs approximate valid stochastic processes. The inconsistency is negligible for moderate context sizes but can be significant in the few-shot regime.

## 1 Introduction

Neural processes (Garnelo et al., 2018a;b) combine the flexibility of neural networks with the uncertainty quantification of Gaussian processes. Given a context set of input-output pairs, a neural process outputs a predictive distribution over outputs at new inputs. This framework has found applications in meta-learning, few-shot prediction, and sequential decision-making.

However, neural processes do not define valid stochastic processes in the sense of Kolmogorov. A stochastic process must satisfy two consistency conditions: marginalization consistency (integrating out variables yields the correct marginal) and conditioning consistency (the chain rule of probability holds). While conditional neural processes (CNPs) satisfy marginalization consistency by construction as their decoder factorizes over target points, they violate conditioning consistency. Adding a point to the context and re-running the encoder does not produce the same result as conditioning a joint distribution.

This violation is well-known. The original neural process paper (Garnelo et al., 2018b) notes that the consistency conditions from the Kolmogorov extension theorem are only approximately satisfied. The Neural Process Family tutorial (Dubois et al., 2020) states explicitly that "in practice, NPs yield good predictive performance even though they can violate this consistency." Recent work on Flow Matching Neural Processes (Hamad & Rosenbaum, 2025) provides detailed discussion of which NP variants satisfy which consistency conditions, noting that CNPs are consistent over marginalization but not conditioning, while other variants like Neural Diffusion Processes (Dutordoir et al., 2023) exhibit the opposite pattern.

Despite this awareness, no prior work has quantified the consistency violation. How large is the gap? Does it vanish as context size grows? Under what conditions is it worst? Without answers to these questions, the statement that NPs "work well despite" inconsistency remains vague.

We provide a quantitative analysis of consistency in neural processes. We define the conditioning consistency gap as a KL divergence measuring how much predictions change when a point is added to the context versus

conditioned upon, and derive exact characterizations and bounds for this quantity. For CNPs with linear decoders and constant variance, we show the gap is $O(1/n^2)$ in context size $n$, and extend this to general Lipschitz decoders where the same rate holds. We further prove this rate is tight by constructing explicit CNPs that achieve it.

These results may explain the empirical success of CNPs. Our analysis also identifies the few-shot regime as where inconsistency is most severe, and characterizes the worst case as arising from "maximally surprising" new observations that differ substantially from the existing context.

## 2 Related Work

The neural process framework was introduced by Garnelo et al. (2018a) for conditional neural processes (CNPs) and extended to latent neural processes (Garnelo et al., 2018b). Subsequent work has developed many variants including attentive neural processes (Kim et al., 2019), convolutional neural processes (Gordon et al., 2020; Foong et al., 2020), and transformer-based neural processes (Nguyen & Grover, 2022). Our analysis focuses on the original CNP architecture.

The consistency issue has been noted since the original papers. Garnelo et al. (2018b) acknowledge that their model only approximately satisfies the Kolmogorov conditions. The Neural Process Family tutorial (Dubois et al., 2020) provides detailed discussion of exchangeability and consistency, noting that different architectures violate different conditions. Recent work (Dutordoir et al., 2023; Hamad & Rosenbaum, 2025) explicitly addresses consistency, with the latter providing a taxonomy of which models satisfy which conditions. However, none of this work quantifies the violation or provides bounds.

There is substantial work on generalization bounds for meta-learning (Baxter, 2000; Pentina & Lampert, 2014; Amit & Meir, 2018; Rothfuss et al., 2021), including information-theoretic approaches (Jose & Simeone, 2021). These bounds address how well a meta-learner generalizes to new tasks, not whether its predictions define a consistent stochastic process. Our work is complementary as we characterize a structural property (consistency) rather than a statistical one (generalization).

## 3 Setup

Let $(\mathcal{X}, \mathcal{Y})$ denote the input and output spaces, with $\mathcal{X} \subseteq \mathbb{R}^{d_x}$ and $\mathcal{Y} \subseteq \mathbb{R}^{d_y}$. A context set is a finite collection $C = \{(x_i, y_i)\}_{i=1}^n \subset \mathcal{X} \times \mathcal{Y}$.

**Definition 1** (Conditional Neural Process). *A Conditional Neural Process (CNP) consists of:*

1. *An encoder $h : \mathcal{X} \times \mathcal{Y} \to \mathbb{R}^d$ mapping context pairs to representations*

2. *An aggregator $a : (\mathbb{R}^d)^n \to \mathbb{R}^d$, typically mean aggregation: $r_C = \frac{1}{n} \sum_{i=1}^n h(x_i, y_i)$*

3. *A decoder with mean function $\mu_\theta : \mathcal{X} \times \mathbb{R}^d \to \mathbb{R}^{d_y}$ and variance function $\sigma_\theta : \mathcal{X} \times \mathbb{R}^d \to \mathbb{R}_{>0}$, defining $p_\theta(y \mid x; C) = \mathcal{N}(\mu_\theta(x, r_C), \sigma_\theta(x, r_C)^2 I)$*

4. *Joint predictions factorize over targets: $p_\theta(y_T \mid x_T; C) = \prod_{t \in T} p_\theta(y_t \mid x_t; C)$*

For a context set $C$ and a new observation $(x_*, y_*)$, we write $C^+ = C \cup \{(x_*, y_*)\}$ for the augmented context.

## 4 Consistency Conditions

A stochastic process defines a consistent family of finite-dimensional distributions satisfying the Kolmogorov extension theorem. For predictive models, this translates to two conditions. For a set of target inputs $x_T = \{x_t\}_{t \in T}$, we write $y_T = \{y_t\}_{t \in T}$ for the corresponding outputs and $p(y_T \mid x_T; C)$ for the joint predictive distribution.

**Definition 2** (Marginalization Consistency). *A predictive model $p(y_T \mid x_T; C)$ is marginalization consistent if for all target sets $T$ and subsets $S \subset T$:*

$$\int p(y_T \mid x_T; C)\, dy_{T \setminus S} = p(y_S \mid x_S; C)$$

**Definition 3** (Conditioning Consistency). *A predictive model is conditioning consistent if for all contexts $C$, observations $(x_*, y_*)$, and targets $(x_\dagger, y_\dagger)$:*

$$p(y_\dagger \mid x_\dagger; C^+) = \frac{p(y_*, y_\dagger \mid x_*, x_\dagger; C)}{p(y_* \mid x_*; C)}$$

*whenever $p(y_* \mid x_*; C) > 0$.*

**Remark 1.** *The factorized form immediately implies marginalization consistency. However, this factorization is what breaks conditioning consistency.*

## 5 The Conditioning Consistency Gap

To quantify the degree to which CNPs violate conditioning consistency, we introduce the following measure.

**Definition 4** (Conditioning Consistency Gap). *For a CNP with context $C$, new observation $(x_*, y_*)$, and target $x_\dagger$, the conditioning consistency gap is:*

$$\Delta(x_*, y_*, x_\dagger; C) = D_{\mathrm{KL}}\left( p_\theta(y_\dagger \mid x_\dagger; C^+) \,\|\, p_\theta(y_\dagger \mid x_\dagger; C) \right)$$

For conditioning consistency to hold, we would require $p(y_\dagger \mid x_\dagger; C^+) = p(y_\dagger \mid x_\dagger; C)$ (since the factorized joint implies the conditional equals the marginal). Thus $\Delta = 0$ iff conditioning consistency holds locally at $(x_*, y_*, x_\dagger)$.

For Gaussian predictive distributions with means $\mu_C, \mu_{C^+}$ and variances $\sigma_C^2, \sigma_{C^+}^2$:

$$\Delta = \log \frac{\sigma_C}{\sigma_{C^+}} + \frac{\sigma_{C^+}^2 + (\mu_{C^+} - \mu_C)^2}{2\sigma_C^2} - \frac{1}{2}$$

**Remark 2** (Consistency vs Usefulness). *The conditioning consistency gap measures deviation from the identity $p(y_\dagger \mid x_\dagger; C^+) = p(y_\dagger \mid x_\dagger; C)$. This identity would hold if the predictive distributions came from a factorized joint $p(y_*, y_\dagger \mid x_*, x_\dagger; C) = p(y_* \mid x_*; C) \cdot p(y_\dagger \mid x_\dagger; C)$, since conditioning on $y_*$ would then provide no information about $y_\dagger$. Ironically, a nonzero gap is what makes CNPs useful since observing $(x_*, y_*)$ should change predictions at $x_\dagger$, and $\Delta = 0$ would imply the model ignores new evidence entirely.*

*The distinction is in the mechanism of the update. A consistent model such as a GP also updates its predictions given new data, but does so through proper conditioning on a well-defined joint distribution. CNPs update predictions by recomputing the representation $r_{C^+}$, which does not correspond to conditioning any joint. The consistency gap quantifies the cost of this shortcut as it measures how far the CNP's update deviates from what any valid conditioning rule could produce.*

## 6 Results

We first establish how the representation changes when augmenting the context.

**Theorem 1** (Consistency Gap for Linear Decoders). *Suppose the decoder has the form $\mu_\theta(x, r) = W(x)^\top r$ for some $W : \mathcal{X} \to \mathbb{R}^{d \times d_y}$, and constant variance $\sigma_\theta(x, r) = \sigma > 0$. Then the conditioning consistency gap satisfies:*

$$\Delta(x_*, y_*, x_\dagger; C) = \frac{\|W(x_\dagger)^\top (h(x_*, y_*) - r_C)\|^2}{2\sigma^2 (n+1)^2}$$

*In particular, if $\|W(x)\|_{\mathrm{op}} \leq B_W$ for all $x$ and $\|h(x, y)\| \leq B_h$ for all $(x, y)$, then:*

$$\Delta(x_*, y_*, x_\dagger; C) \leq \frac{2 B_W^2 B_h^2}{\sigma^2 (n+1)^2} = O\left(\frac{1}{n^2}\right)$$

*Proof.* Under mean aggregation, adding $(x_*, y_*)$ to a context of size $n$ gives $r_{C+} = (n \cdot r_C + h(x_*, y_*))/(n+1)$, so:

$$r_{C+} - r_C = \frac{h(x_*, y_*) - r_C}{n+1}$$

With constant variance $\sigma_{C+} = \sigma_C = \sigma$, the KL divergence simplifies to:

$$\Delta = \frac{(\mu_{C+} - \mu_C)^2}{2\sigma^2}$$

The mean difference is:

$$\mu_{C+}(x_\dagger) - \mu_C(x_\dagger) = W(x_\dagger)^\top (r_{C+} - r_C) = \frac{W(x_\dagger)^\top (h(x_*, y_*) - r_C)}{n+1}$$

Substituting yields the first claim. For the bound, note that $\|r_C\| = \|\frac{1}{n}\sum_i h(x_i, y_i)\| \leq \frac{1}{n}\sum_i \|h(x_i, y_i)\| \leq B_h$ by the triangle inequality, so $\|h(x_*, y_*) - r_C\| \leq 2B_h$. $\square$

The bound in Theorem 1 gives a criterion for when the gap becomes negligible.

**Corollary 1** (Vanishing Gap). *Under the conditions of Theorem 1, for any $\epsilon > 0$, the conditioning consistency gap satisfies $\Delta < \epsilon$ whenever:*

$$n > \sqrt{\frac{2B_W^2 B_h^2}{\sigma^2 \epsilon}} - 1$$

**Remark 3** (Decoder Parameterization). *Theorem 1 uses the parameterization $\mu_\theta(x, r) = W(x)^\top r$, where the weight depends on the target input. An alternative is the concatenation form $\mu_\theta(x, r) = W^\top [r; x]$, common in conditional generative models. The results are identical since $x_\dagger$ is fixed when comparing $\mu_{C+}$ and $\mu_C$, the mean difference depends only on $\delta_r$, with the submatrix $W_r$ playing the role of $W(x_\dagger)$. The concatenation form is slightly simpler since the effective weight matrix is independent of $x$, making the bound uniform over target locations automatically.*

We now extend to general nonlinear decoders under a Lipschitz condition. The extension to nonlinear decoders requires understanding how the KL divergence behaves under small perturbations to both mean and variance. The following lemma shows it is locally quadratic.

**Lemma 1** (KL Divergence Locally Quadratic). *For Gaussians with $\mu_1 = \mu_0 + \epsilon_\mu$ and $\sigma_1 = \sigma_0 + \epsilon_\sigma$ where $|\epsilon_\sigma| < \sigma_0/2$:*

$$D_{\mathrm{KL}}(\mathcal{N}(\mu_1, \sigma_1^2)\|\mathcal{N}(\mu_0, \sigma_0^2)) = \frac{\epsilon_\mu^2}{2\sigma_0^2} + \frac{\epsilon_\sigma^2}{\sigma_0^2} + O(\epsilon^3)$$

*where $\epsilon = \max(|\epsilon_\mu|, |\epsilon_\sigma|)$.*

*Proof.* Expand the KL divergence:

$$D_{\mathrm{KL}} = \log \frac{\sigma_0}{\sigma_1} + \frac{\sigma_1^2 + (\mu_1 - \mu_0)^2}{2\sigma_0^2} - \frac{1}{2}$$

Using $\log(1+x) = x - x^2/2 + O(x^3)$:

$$\log \frac{\sigma_0}{\sigma_0 + \epsilon_\sigma} = -\frac{\epsilon_\sigma}{\sigma_0} + \frac{\epsilon_\sigma^2}{2\sigma_0^2} + O(\epsilon_\sigma^3)$$

Expanding the variance ratio:

$$\frac{(\sigma_0 + \epsilon_\sigma)^2}{2\sigma_0^2} = \frac{1}{2} + \frac{\epsilon_\sigma}{\sigma_0} + \frac{\epsilon_\sigma^2}{2\sigma_0^2}$$

Combining, the linear terms in $\epsilon_\sigma/\sigma_0$ cancel, yielding:

$$D_{\mathrm{KL}} = \frac{\epsilon_\sigma^2}{\sigma_0^2} + \frac{\epsilon_\mu^2}{2\sigma_0^2} + O(\epsilon^3)$$

$\square$

**Theorem 2** (Consistency Gap for Lipschitz Decoders). *Suppose $\mu_\theta(x, \cdot)$ is $L_\mu$-Lipschitz and $\sigma_\theta(x, \cdot)$ is $L_\sigma$-Lipschitz for all $x$, with $\sigma_\theta(x, r) \geq \sigma_{\min} > 0$. If $\|h(x, y)\| \leq B_h$ for all $(x, y)$, then for sufficiently large $n$:*

$$\Delta(x_*, y_*, x_\dagger; C) \leq \frac{2(L_\mu^2 + 2L_\sigma^2)B_h^2}{\sigma_{\min}^2(n+1)^2} + O\left(\frac{1}{n^3}\right) \quad \text{for } n > \frac{4L_\sigma B_h}{\sigma_{\min}}$$

*Proof.* Let $\delta_r = r_{C^+} - r_C$, so $\|\delta_r\| \leq 2B_h/(n+1)$ (as in the proof of Theorem 1). By Lipschitz continuity:

$$|\mu_{C^+} - \mu_C| \leq L_\mu \|\delta_r\| \leq \frac{2L_\mu B_h}{n+1}$$
$$|\sigma_{C^+} - \sigma_C| \leq L_\sigma \|\delta_r\| \leq \frac{2L_\sigma B_h}{n+1}$$

The KL divergence between Gaussians $\mathcal{N}(\mu_1, \sigma_1^2)$ and $\mathcal{N}(\mu_0, \sigma_0^2)$ with $\mu_1 = \mu_0 + \epsilon_\mu$ and $\sigma_1 = \sigma_0 + \epsilon_\sigma$ satisfies:

$$D_{\mathrm{KL}} = \frac{\epsilon_\mu^2}{2\sigma_0^2} + \frac{\epsilon_\sigma^2}{\sigma_0^2} + O(\epsilon^3)$$

Substituting $\epsilon_\mu \leq 2L_\mu B_h/(n+1)$, $\epsilon_\sigma \leq 2L_\sigma B_h/(n+1)$, and $\sigma_0 \geq \sigma_{\min}$ yields the result. $\square$

**Remark 4.** *Unlike typical perturbation bounds that are $O(\epsilon)$, the KL divergence between nearby Gaussians is $O(\epsilon^2)$ because the linear terms cancel. This is a manifestation of the fact that KL divergence is locally quadratic, related to Fisher information.*

## 7 Tightness of Bounds

We now show that the $O(1/n^2)$ rate in Theorem 1 is tight and there exist CNPs achieving this rate.

**Theorem 3** (Lower Bound). *For any $B_W, B_h, \sigma > 0$, there exists a CNP with $\|W(x)\|_{\mathrm{op}} \leq B_W$, $\|h(x, y)\| \leq B_h$, and constant variance $\sigma$, and a sequence of contexts $C_n$ of size $n$, such that:*

$$\Delta(x_*, y_*, x_\dagger; C_n) = \frac{2B_W^2 B_h^2}{\sigma^2(n+1)^2}$$

*matching the upper bound in Theorem 1.*

*Proof.* We construct an explicit example. Let $d = d_y = 1$ (scalar representations and outputs) and define:

- Encoder: $h(x, y) = B_h \cdot \mathrm{sign}(y) \cdot \mathbf{1}_{|y|>0}$, i.e., $h$ outputs $+B_h$ for positive $y$ and $-B_h$ for negative $y$

- Decoder mean: $\mu_\theta(x, r) = B_W \cdot r$ (linear with slope $B_W$)

- Decoder variance: $\sigma_\theta(x, r) = \sigma$ (constant)

Consider the context $C_n = \{(x_i, y_i)\}_{i=1}^n$ where all $y_i < 0$. Then:

$$r_{C_n} = \frac{1}{n}\sum_{i=1}^n h(x_i, y_i) = \frac{1}{n} \cdot n \cdot (-B_h) = -B_h$$

Now take the new observation $(x_*, y_*)$ with $y_* > 0$, so $h(x_*, y_*) = +B_h$. The representation difference is:

$$h(x_*, y_*) - r_{C_n} = B_h - (-B_h) = 2B_h$$

By Theorem 1, the consistency gap is:

$$\Delta = \frac{|W(x_\dagger)|^2 \cdot |h(x_*, y_*) - r_C|^2}{2\sigma^2(n+1)^2} = \frac{B_W^2 \cdot 4B_h^2}{2\sigma^2(n+1)^2} = \frac{2B_W^2 B_h^2}{\sigma^2(n+1)^2}$$

$\square$

**Remark 5** (Interpretation). *The lower bound is achieved when:*

1. *The new observation is "maximally surprising" relative to the context (opposite sign of all context points)*

2. *The decoder fully utilizes its capacity (weight at the bound $B_W$)*

*In benign cases where new observations are similar to the context, the gap will be much smaller. The $O(1/n^2)$ rate represents the worst case.*

**Theorem 4** (Tightness for Lipschitz Decoders). *For any $L_\mu, L_\sigma, B_h, \sigma_{\min} > 0$, there exists a CNP with $L_\mu$-Lipschitz mean decoder, $L_\sigma$-Lipschitz variance decoder, $\sigma_\theta \geq \sigma_{\min}$, $\|h(x,y)\| \leq B_h$, and a sequence of contexts $C_n$ of size $n$, such that:*

$$\Delta(x_*, y_*, x_\dagger; C_n) = \frac{2(L_\mu^2 + 2L_\sigma^2)B_h^2}{\sigma_{\min}^2(n+1)^2} + O\left(\frac{1}{n^3}\right)$$

*matching the leading constant in Theorem 2.*

*Proof.* Let $d = d_y = 1$ and define:

- Encoder: $h(x,y) = B_h \cdot \text{sign}(y)$

- Decoder mean: $\mu_\theta(x,r) = L_\mu \cdot r$      ($L_\mu$-Lipschitz, tight everywhere)

- Decoder variance: $\sigma_\theta(x,r) = \sigma_{\min} + L_\sigma|r - r_0|$ for a reference point $r_0$

Consider contexts $C_n$ with all $y_i < 0$, so $r_{C_n} = -B_h$, and set $r_0 = -B_h$. Take $(x_*, y_*)$ with $y_* > 0$. Then $\delta_r = 2B_h/(n+1)$ and:

$$\epsilon_\mu = L_\mu \cdot \delta_r = \frac{2L_\mu B_h}{n+1}$$
$$\epsilon_\sigma = L_\sigma \cdot \delta_r = \frac{2L_\sigma B_h}{n+1}$$
$$\sigma(r_{C_n}) = \sigma_{\min}$$

All three inequalities in the proof of Theorem 2 are tight simultaneously. Applying Lemma 1:

$$\Delta = \frac{\epsilon_\mu^2}{2\sigma_{\min}^2} + \frac{\epsilon_\sigma^2}{\sigma_{\min}^2} + O(\epsilon^3) = \frac{2(L_\mu^2 + 2L_\sigma^2)B_h^2}{\sigma_{\min}^2(n+1)^2} + O\left(\frac{1}{n^3}\right)$$

$\square$

**Remark 6** (Role of Variance Dependence). *One might expect that allowing variance to depend on the representation would worsen the consistency gap. However, the $O(1/n^2)$ rate holds regardless. This follows from that the KL divergence between nearby Gaussians is locally quadratic in both mean and variance perturbations, with the linear terms cancelling. Both $\epsilon_\mu$ and $\epsilon_\sigma$ are $O(1/n)$, so both contribute at $O(1/n^2)$.*

## 8 Numerical Experiments

We verify our theoretical results and investigate the role of the Lipschitz condition through numerical experiments. All experiments use scalar representations ($d = 1$), context sizes $n = 2$ to $300$, and $300$ random trials per $n$.

Figure 1(a) confirms Theorems 1 and 3: the worst-case construction matches the upper bound exactly, while random contexts produce gaps well below the bound. Figure 1(b) shows worst-case gaps for six nonlinear Lipschitz decoders (tanh, sinusoidal, ReLU, ELU with sigmoid variance, cubic, and log-contractive). All

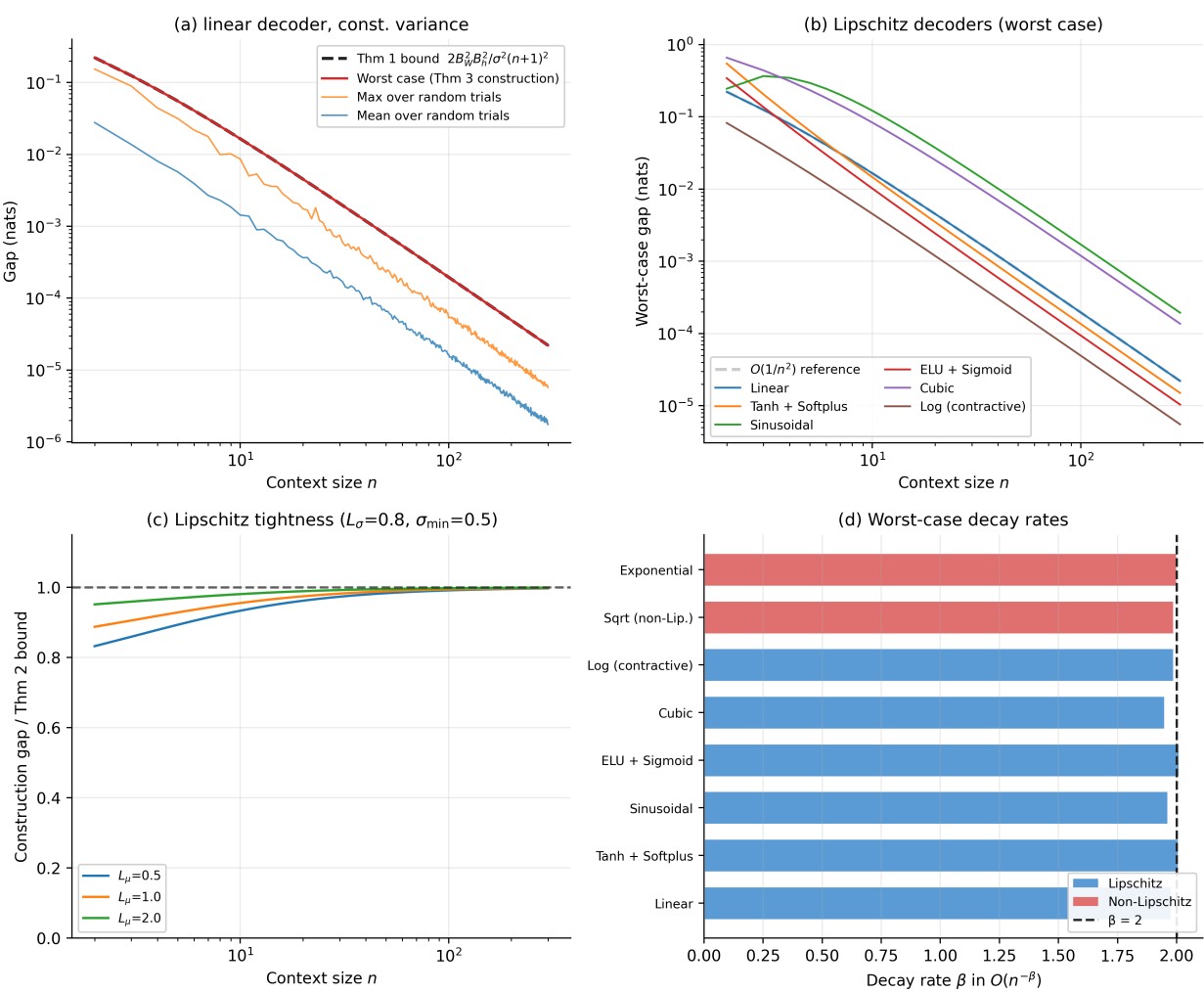

Figure 1: Numerical validation of the theoretical results.

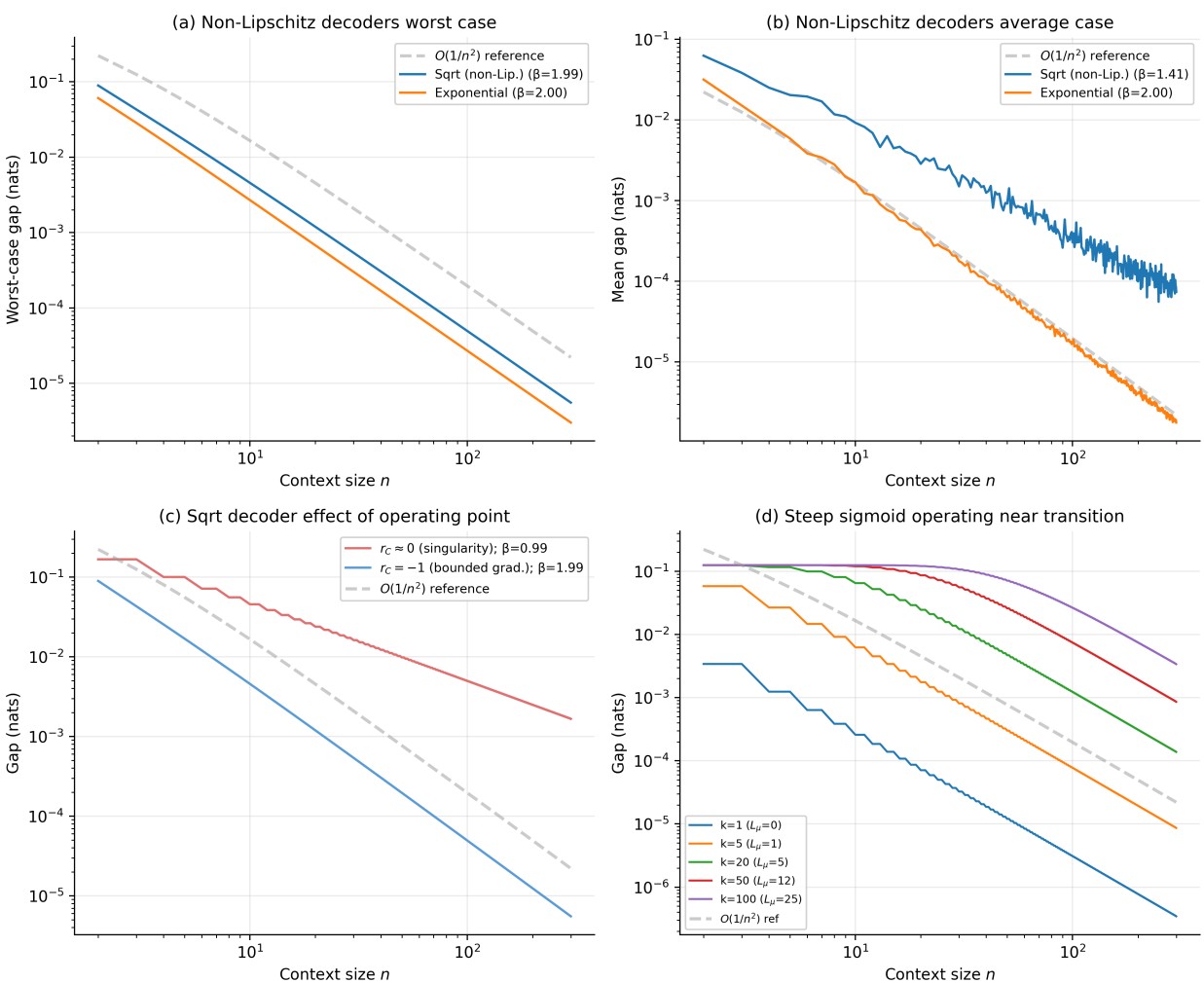

Figure 2: Consistency gap for non-Lipschitz and boundary-case decoders.

exhibit decay rate $\beta \approx 2.0$ when fitting $\Delta \propto n^{-\beta}$, confirming the $O(1/n^2)$ rate of Theorem 2. Figure 1(c) verifies the tightness of Theorem 4 that the ratio of the construction's gap to the bound converges to 1 as $n$ grows.

Figure 2 investigates decoders that violate or stress the Lipschitz condition. The sqrt decoder ($\mu(r) = \text{sign}(r)\sqrt{|r|}$, with unbounded gradient at the origin) achieves $\beta \approx 2.0$ in worst case (Figure 2(a)), because mean aggregation ensures $\|\delta_r\| = O(1/n)$ and the decoder's local Lipschitz constant at the operating point is what governs the gap. When $r_C \approx 0$ (at the singularity), the rate degrades to $\beta \approx 1.0$ (Figure 2(c)), but this requires the representation to sit exactly at the singularity. The exponential decoder ($\mu(r) = e^r$, not globally Lipschitz) also achieves $\beta \approx 2.0$, since it is Lipschitz on any bounded domain.

The steep sigmoid decoder ($\mu(r) = \text{sigmoid}(kr)$, Lipschitz with constant $k/4$) maintains $O(1/n^2)$ rate but with a constant proportional to $k^2$ (Figure 2(d)). For large $k$, the bound is uninformative at small $n$.

These experiments suggest that Lipschitzness is sufficient but not necessary for $O(1/n^2)$; local Lipschitzness at the operating point suffices, which holds for any smooth decoder on a bounded representation space. The load-bearing assumption is $\sigma_{\min} > 0$, not Lipschitzness of the mean decoder, since standard variance parameterizations ($\text{softplus}(\cdot) + \varepsilon$, $\exp(\cdot)$) enforce positive variance in practice.

## 9  Discussion

Our results establish that the conditioning consistency gap for CNPs with bounded encoders and Lipschitz decoders is $O(1/n^2)$ in context size, and that this rate is tight. This provides a theoretical explanation for the empirical observation that CNPs work well in practice despite not defining valid stochastic processes, because the inconsistency becomes negligible for sufficiently large context sets. For a context of size $n = 100$ and reasonable constants ($B_W = B_h = 1$, $\sigma = 1$), the consistency gap is on the order of $10^{-4}$ nats, which is unlikely to affect downstream performance in most applications.

The $O(1/n^2)$ rate has practical implications. Consistency violations are most severe in the few-shot regime with fewer than ten context points, which is a regime where CNPs are often deployed. For moderate context sizes above fifty points, CNPs are approximately consistent and the gap becomes negligible. Applications requiring strict consistency, such as certain sequential decision making settings where predictions must cohere across time, should either ensure sufficiently large contexts or consider alternative architectures that guarantee consistency by construction.

An interesting consequence of our analysis is that the $O(1/n^2)$ rate holds regardless of whether the decoder variance depends on the representation. One might expect that allowing variance to change with context would introduce additional inconsistency, but Lemma 1 shows that the KL divergence between nearby Gaussians is locally quadratic in both mean and variance perturbations. The linear terms cancel, yielding the same asymptotic rate.

Our results use the closed-form KL for Gaussians, but the local quadratic structure of KL divergence holds generally via the Fisher information metric, suggesting the $O(1/n^2)$ rate extends to other exponential family decoders. For non-exponential family decoders, the local geometry of KL may differ and the rate remains an open question.

Several questions remain open. How does the gap behave for latent neural processes with stochastic encoders? The latent variable introduces additional structure that may either help or hurt consistency. What is the relationship between the conditioning consistency gap and downstream task performance? Our bounds are information-theoretic, but the operational consequences for tasks like Bayesian optimization are unclear. Can training objectives be modified to explicitly minimize the consistency gap, perhaps through a regularizer that penalizes large representation shifts?

Finally, do similar bounds hold for attention-based aggregation, which are increasingly popular in modern neural process variants? Attention does not satisfy the simple update rule, but we leave this to future work.

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
