# OpenReview forum: "On the Conditioning Consistency Gap in Conditional Neural Processes"
_TMLR — Accepted by TMLR_

### Review · Reviewer_cWtx · 2026-02-19

**Summary Of Contributions:**

Conditional neural processes (CNPs) are trained to do in-context learning---given a context set $C$ of $(x, y)$ pairs, it outputs the predictive distribution $p(y_* | x_*, C)$ at a new location $x_*$.
They tend to work well in practice, but it is puzzling as to why since CNPs are not conditioning consistent.
This means, unlike Gaussian processes, CNPs can "revise" their predictions on previously observed data, given additional conditioning data.
Intuitively, once the truth has been uncovered, a conditioning consistent process should not be influenced by additional observations.

This paper defines the conditioning consistency gap to quantify the above.
Using a "toy" setting of linear decoders, the authors show that this gap is upper-bounded by $O(n^{-2})$ in the context size $n$.
Thus, linear CNPs' gap will vanish for a sufficiently large context set.
Moreover, this bound is tight, i.e., there exists a linear CNP that realizes this bound.
The authors show the same bound for Lipschitz decoders, although no tightness result is derived.

**Audience:**

Yes

**Audience Explanation:**

The results in this paper provide a first step in understanding CNPs and potentially various other in-context, amortized prediction models (e.g. TabPFN, TabICL, etc.).

**Claims And Evidence:**

Yes

**Claims Explanation:**

All proofs are provided inline and accessible.
As far as I see, they are correct.

**Requested Changes:**

1. Add numerical experiments and discussions to validate the theoretical results, esp. for Thm 2 & 3. This is particularly needed since it is unclear if the bound is tight.
2. Add experiments on more general decoder classes. This is useful to check whether the Lipschitzness condition is limiting. Provide additional discussion on this, too.
3. Any experiments will strengthen the paper greatly.

In my opinion, both 1 and 2 are essential for acceptance.
It doesn't mean that bad results on more general assumptions on CNPs are grounds for rejection.
Rather, I'm suggesting them so that the paper will be much more complete, useful, and insightful.

---

> ### Author Response · Authors · 2026-02-20
>
> We thank the reviewer for the feedback. We conducted both requested experiments and summarize the findings below. These will be incorporated into the revised draft as figures.
>
> We numerically verified Theorems 1–3 across context sizes $n = 2$ to $300$ with 300 random trials per $n$.
>
> - The worst-case construction from Theorem 3 matches the upper bound $\frac{2B_W^2 B_h^2}{\sigma^2(n+1)^2}$ exactly across all $n$.
> - We tested six nonlinear Lipschitz decoders: $\tanh$, sinusoidal, ReLU, ELU + sigmoid variance, cubic polynomial, and log-contractive. All exhibit worst-case decay rate $\beta \approx 2.0$ when fitting $\Delta \propto n^{-\beta}$, confirming the $O(1/n^2)$ rate.
> - We show tightness for Theorem 2 with an explicit construction $\mu(r) = L_\mu \cdot r$ with $\sigma(r) = \sigma_{\min} + L_\sigma |r - r_C|$. Under the standard worst case ($r_C = -B_h$, $h(x_*, y_*) = +B_h$), the gap evaluates to
>
> $$\Delta = \frac{\varepsilon_\mu^2}{2\sigma_{\min}^2} + \frac{\varepsilon_\sigma^2}{\sigma_{\min}^2} + O(1/n^3) = \frac{2(L_\mu^2 + 2L_\sigma^2)B_h^2}{\sigma_{\min}^2(n+1)^2} + O(1/n^3)$$
>
> matching the leading constant. Numerically, the ratio of the construction's gap to the bound converges to 1 as $n$ grows.
>
> We tested several non-Lipschitz decoders to determine whether the Lipschitz condition is limiting, including sqrt (unbounded gradient at origin), exponential (not globally Lipschitz), and steep sigmoid (Lipschitz but with large constant). we find:
>
> - Sqrt and exponential both achieve $\beta \approx 2.0$ in worst case, showing Lipschitzness is sufficient but not necessary. Local Lipschitzness at the operating point suffices, which holds for any smooth decoder on a bounded representation space.
> - The steep sigmoid ($k = 100$, $L_\mu = 25$) maintains $O(1/n^2)$ rate but with a constant proportional to $k^2$, making the bound vacuous for small $n$; the Lipschitz constant's magnitude matters more than whether the condition holds.
> - The load bearing assumption is $\sigma_{\min} > 0$ and not Lipschitzness of the mean decoder. Variance collapse can amplify the gap arbitrarily, but this does not arise in practice since standard decoder parameterizations ($\text{softplus}(\cdot) + \varepsilon$, $\exp(\cdot)$) enforce positive variance.

---

> > ### Comment · Reviewer_cWtx · 2026-02-20
> >
> > Thank you for the experiments!
> >
> > I just have one additional question: Why does the linear decoder in Thm. 1 have the form $W(x)^\top r$, i.e., the weight depends on $x$? In conditional generative models (like VAE), usually one would do something like $W [r, x]$ instead, i.e., the conditioning variable is concatenated with the latent, and the weight is independent of them.

---

> > > ### Author Response · Authors · 2026-02-21
> > >
> > > Thanks for the question. The parameterization was a modeling choice (used in FiLM and kernel methods for example) but the results are the same since $x^\dagger$ is fixed when comparing $\mu_{C^+}$ and $\mu_C$, the mean difference depends only on $\delta r$, and if $f_\theta$ is Lipschitz in its first argument, the $O(1/n^2)$ rate follows from Theorem 2. We will add a remark to the paper noting both parameterizations and their equivalence for our results.

---

### Review · Reviewer_N53P · 2026-03-04

**Summary Of Contributions:**

Neural processes (NPs) map contexts to predictive distributions, and are trained via meta-learning where one provides contexts labelled with predictions. NPs are inspired by stochastic processes but do not satisfy the Kolmogorov consistency (i.e. conditioning is consistent), which are required for SPs. The authors define a consistency gap which measures the degree to which NPs do not satisfy consistency. They show that NPs with bounded encoders / Lipschitz decoders, the consistency gap decreases like 1/n^2 in context size n. Thus, CPs are effectively valid SPs for moderate context sizes, but not in a few-shot setting.

**Audience:**

No

**Audience Explanation:**

I do not click "No" without hesitation.

I do believe that the topic studied is of interest to some individuals, but the findings in this paper are extremely light and therefore perhaps not of enough interest to warrant a full paper. The proof of Theorem 1 / 2 just rely on the closed-form wasserstein distance for Gaussians and Lipschitzness / boundedness. The proof is understandably very short. No experiments are provided to give insight into how the result manifests empirically. No distributions beyond Gaussian are considered.

**Broader Impact Concerns:**

No broader impact statement required for this type of work.

**Claims And Evidence:**

Yes

**Claims Explanation:**

Strengths:
- Overall, most of the results in this paper seem correct, although they are presented in a very awkward manner.

Weaknesses:
- I feel as though the notation/definitions could be a bit more explicit between section 3 and 4. For instance, in definition 2, the target set $T$ is defined, but the variable $y_T$ is not defined. I assume this is a (multivariate) joint distribution over predictions on a target set. Similarly, in Remark 1, this seems to be a definition rather than a result. If it is a definition, why isn't it stated in Definition 1, which seems to not explicitly state that joint predictions are made via a factorised model?
- Similar to the above, I think in general more words between the labelled environments (Definition, remark, lemma) would be helpful to orient the reader and give extra context.
- The mean and variance functions in Definition 1.3 are not defined (in particular, their domain and ranges). It is important because the KL divergence in 4 would require a positive definite covariance.
- Lemma 1 is trivial and I'm not sure it is necessary to include in the main paper (certainly not the proof). I would suggest using an appendix.
- Proof of theorem 1 probably could go to the appendix as well.
- In proof of theorem 1, it is stated that the norm of $r_C$ is less than something by convexity. But isn't this actually not convexity, but the fact that $h$ was bounded in the assumptions of the theorem?
- Section 6 and 7 could be merged as one section, as the titles are a bit awkward at the moment.
- There is something very strange about the presentation of Lemma 2, Theorem 2, and the proof of Theorem 2. **I do suspect that AI was used to generate significant parts of this paper, and while that is not a problem in itself, the paper should be consolidated by a human who understands the technical elements of the paper.**
    - The proof of Theorem 2 uses Lemma 2, which is fine, but then typically Lemma 2 would appear in an appendix rather than the main paper.
    - The title of section 8.1 is awkward. Why is this a ``corrected bound'' --- is the previous bound not correct?
    - It is stated that "Theorem 2 can be strengthened". But the rate presented in Corollary 2 appears to be the same as in Theorem 2. In fact theorem 2 already uses Lemma 2 in the proof, which already accounts for the Lipschitzness, so how could it be a strengthening??
    - The paper is quite light: Many of the results / proofs should be in an appendix. Once this happens, the length of the paper becomes much shorter. No experiments are present, which I feel could bolster the paper. For instance, one could measure the scenarios described in the discussion empirically.

Minor comments:
- CNP (conditional neural process) is used many times before it is defined in section 2. It also appears in the abstract.

**Requested Changes:**

Overall, I think the paper could be modified to satisfy the correctness criterion by fixing the presentation issues (as described above).
Adding experiments would also be required to satisfy the correctness criterion.
Adding experiments will contribute towards satisfying the "interest" criterion, but it will still be quite a light paper.

---

> ### Author Response · Authors · 2026-03-05
>
> Thank you for the detailed feedback. We have revised the manuscript to address each point.
>
> Lemma 2 now appears before Theorem 2 and some sections have been removed or merged as suggested. Definition 1 states the factorized prediction form and specifies the domain and range of the mean and variance functions. CNP is defined on first use in the abstract. The target set notation $y_T$ is introduced before definition 2. The bound in the proof of Theorem 1 now references the triangle inequality. We have added connective text between formal environments throughout.
>
> We have added a new theorem which constructs a CNP achieving the leading constant of Theorem 2 asymptotically.
>
> We also added a numerical experiments section with two figures. Figure 1 validates Theorems 1–3 and the Lipschitz tightness result across a few nonlinear Lipschitz decoders. Figure 2 tests non-Lipschitz decoders (sqrt, exponential, steep sigmoid) and finds that Lipschitzness is sufficient but not necessary; local Lipschitzness at the operating point suffices and the load bearing assumption is $\sigma_{\min} > 0$.
>
> We prefer to keep proofs in the main text since there is no page limit and inline proofs are easier to follow.
>
> Extending to non Gaussian predictive families where the KL divergence lacks a closed form is an open question. We have added this to the discussion.

---

> > ### Comment · Reviewer_N53P · 2026-03-18
> >
> > Thanks for uploading the new manuscript with technical fixes. In light of these changes, I have updated my response of "Are the claims made in the submission supported by accurate, convincing and clear evidence?" to "Yes".
> >
> > Regarding non-Gaussian predictive families with closed-form KL divergences --- there are many (see named exponential families). But this is adjacent to my concern, which is not about closed-form KL divergence, but about the broad applicability of the results. That is, it is possible to have more results without having closed-form KL divergences.
> >
> > I maintain that this paper is not of interest to at least some individuals in TMLR's audience, because the findings are too light.

---

### Review · Reviewer_ho9n · 2026-03-05

**Summary Of Contributions:**

The authors address the widely acknowledged but previously unquantified inconsistency in Conditional Neural Processes (CNPs) by formally defining the "conditioning consistency gap" as a KL divergence. They prove that for CNPs with bounded encoders and Lipschitz decoders, this gap shrinks at a rate of O(1/n^{2}) with respect to the context size n. They establish that this bound is tight by constructing specific worst-case scenarios involving "maximally surprising" observations. Their findings provide theoretical justification for why CNPs perform well empirically: the consistency violations are most severe in the few-shot regime but become practically negligible for moderate-to-large context sizes.

For a top CS conference venue I would have wanted to see some numerical experiments supporting that the theory plays out in practice, but for TMLR I think the theory alone is sufficient.

**Audience:**

Yes

**Audience Explanation:**

(C)NP are of broad interest, and there is still a limited understanding of their theoretical properties.

**Claims And Evidence:**

Yes

**Claims Explanation:**

This is a purely theoretical paper so the results are readily verifiable.

**Requested Changes:**

I'm not sure Lemma 1 should be labeled as such - it feels somewhat trivial.
Define CNP.

---

> ### Author Response · Authors · 2026-03-05
>
> Thank you for the review. We have rolled Lemma 1 into the proof of Theorem 1 and defined CNP on first use. The revised manuscript also includes numerical experiments and a tightness result for the Lipschitz case.

---

### Decision · Action_Editor_Cz8i · 2026-04-13

**Recommendation:** Accept as is

**Audience:**

Yes

**Audience Explanation:**

Two of the three reviewers consider the work to be of interest to some parts of TMLR's readership.
While I agree with reviewer N53P that the theory is relatively light, I follow the other two in their argument that there is still enough novelty for it to fulfill the interest constraint.

**Claims And Evidence:**

Yes

**Claims Explanation:**

The paper defines a KL-based _consistency gap_ for conditional neural processes (CNPs) and proves that it decays at $\mathcal O(1/n^2)$ (under bounded encoders and Lipschitz decoders), quantifying a gap between CNPs and stochastic processes.
All three reviewers agree that the proofs are correct. The revisions added numerical experiments that numerically confirm the predicted rates.

---

> ### Author Response · Authors · 2026-04-16
>
> We thank the AE and all three reviewers for their constructive feedback and thoughtful evaluation throughout the review process. We have uploaded the camera ready version. Thank you for your time and effort.